# Daily Gain, Feed Conversion, and Rumen Fermentation in Finishing Steers Fed a Total Mixed Ration Supplemented with a Blend of Essential Oils, Tannins, and Bioflavonoids or Monensin [note 1]

**DOI:** 10.3390/ani15040594

**Published:** 2025-02-18

**Authors:** José Luis Repetto, Eliana Ciancio, Guillermo Castro, Alvaro Santana, Cecilia Cajarville

**Affiliations:** Departamento de Producción Animal y Salud de Sistemas Productivos (IPAV), Facultad de Veterinaria, Universidad de la República O. del Uruguay (UdelaR), Ruta 1, km 42.2, San José 8100, Uruguay; joselorepetto@gmail.com (J.L.R.); ciancio232@gmail.com (E.C.); guillermocastro181@gmail.com (G.C.); bobasantana@gmail.com (A.S.)

**Keywords:** beef, natural additives, monensin, essential oils, tannins, bioflavonoids

## Abstract

This study examined how adding a combination of essential oils, tannins, and bioflavonoids, along with monensin, affects the performance and rumen fermentation of finishing steers fed a mixed diet. Thirty steers, each weighing about 441 kg and averaging 34 months of age, were divided into three groups based on the supplements they received. The first group was given an essential oil tannin blend, the second received monensin, and the third received both. Over a 60-day trial, which included a 19-day adjustment period, daily feed intake, weight gain, feed efficiency, and rumen fermentation characteristics such as pH, ammonia nitrogen, and volatile fatty acids were measured. The results showed that the group receiving the essential oils, tannins, and bioflavonoids blend had higher feed intake than the combination group. However, there were no significant differences in weight gain or feed efficiency across the groups. While the rumen pH and ammonia levels remained constant, the combination of treatments resulted in more stable levels of volatile fatty acids, indicating potential benefits from the combined supplementation, which warrants further investigation.

## 1. Introduction

The rise in global beef meat demand imposes the need for a sustainable intensification of the production systems. Intensive beef production often faces challenges in optimizing both animal performance, animal welfare and health, while reducing environmental impact. It is well known that in intensive meat production systems, good rumen functioning is directly associated with animal health. Due to the type of diet used in these systems, with very high proportions of concentrate, rumen health deserves special care to guarantee high gains and conversion efficiencies [1]. One of the most used additives for this purpose is ionophores, carboxylic polyether antibiotics naturally produced by *Streptomyces* spp. Particularly, monensin, an ionophor produced by *Streptomyces cinnamonensis*, is widely used and studied [2]. This antibiotic acts mainly by reducing the cell viability of Gram-positive bacteria and protozoa [3] and has demonstrated positive effects on performance and conversion in beef cattle [4]. Regarding ruminal parameters, de Moura et al. [5], in their review of 52 peer-reviewed publications, reported that the administration of monensin does not affect NH_3_-N concentration but alters the volatile fatty acids (VFAs) profile by increasing propionate and decreasing the butyrate concentration. The widespread use of monensin as a growth promoter in Europe ended in January 2006, when the regulation 1831/2003 of the European Parliament and of the Council (https://eur-lex.europa.eu/legal-content/EN/TXT/PDF/?uri=CELEX:32003R1831 (accessed on 10 January 2025)) became effective. This regulation prevented the use of antibiotics as growth-promoting agents in diets and led to the search for alternatives, mainly through plant-derived compounds and extracts [6]. However, due to the worldwide accepted action, even in Europe, monensin has continued to be used as a positive control in experiments [7].

Essential oils (EOs) are plant-derived compounds widely used instead of monensin, and, according to several reports, could represent a good alternative [8,9]. However, according to others, monensin showed clearer effects compared to EOs, e.g., improving the feed efficiency and energy status of dairy cows [10]. Adding to this, there is very limited information regarding the synergy between essential oils and other phytogenic compounds, such as tannins, which could potentially enhance the effectiveness of the blend.

ANAVRIN^®^ (ANAVRIN^®^, VetosEurope, Lugano, Swuitzerland) contains EOs, and also includes tannins and bioflavonoids, which can improve the action of the additive. The EOs are provided by cloves (*Syzygium aromaticum*), coriander seeds (*Coriandrum sativum*), and geranium (*Pelargonium cucullatum*). Cloves contain eugenol (4-allyl-2-methoxyphenol), known for its antimicrobial properties against both Gram-positive and Gram-negative bacteria [11]. Geranium oils also provide antibacterial, with additional antifungal [12] and antioxidant effects [13]. Coriander oils positively impacted digestibility in ruminants by modulating rumen fermentation [14] and have recognized antioxidant activities and reducing power [15]. ANAVRIN^®^ also contains tannins from chestnuts (*Castanea sativa*), with beneficial properties on protein metabolism and anti-inflammatory action [16,17], as well as bioflavonoids derived from olives (*Olea europaea*), which have antioxidants, anti-inflammatory, and antimicrobial properties [18]. Recent studies communicated that ANAVRIN^®^ increased the productivity of dairy cows [19] and beef calves [20], in experiments that compared its incorporation against control diets without additives. Comparative studies are required to assess ANAVRIN’s efficacy across diverse production systems and conditions. Specifically, the evaluation of the activity of this specific blend, which includes tannins and bioflavonoids in addition to EOs, has not previously been tested with monensin. It is necessary then, to evaluate its performance relative to monensin as a potential antibiotic replacement.

It is to note that monensin is not banned in several regions (e.g., North and South America). Considering that these are the regions with the highest beef production in the world [21], it would be interesting to evaluate whether the mixture of both additives has complementary effects, due to their similar mechanisms of action and expected outcomes. The mechanisms of action, however, have not been sufficiently studied for phytogenic mixtures [22,23], among which is the additive under study.

This study aimed to provide information on the efficacy of a blend of essential oils, tannins, and bioflavonoids compared to monensin, while also investigating any potential additive effects from their combined use on daily gain, feed efficiency, and rumen fermentation parameters of finishing steers consuming a total mixed ration.

## 2. Materials and Methods

### 2.1. Experimental Design and Feeding Management

The experiment was conducted at the IPAV, Faculty of Veterinary Science, Universidad de la República (UdelaR), Uruguay (Ruta 1 km 42.5, San José, Uruguay; GPS coordinates −34.68486892825304, −56.541669330312466). Animal procedures and management adhered to the guidelines Animal Experimentation Ethics Committee of the UdelaR (Protocol: 11982021). Thirty crossbred beef steers (441 ± 27 kg body weight (BW), 34.3 ± 7.1 months old) consuming a TMR were blocked by BW and randomly assigned to three dietary treatments (n = 10 per treatment), which consisted in the addition to the TMR of (1) EOTB (a blend of essential oils, tannins, and bioflavonoids), (2) MON (monensin), and (3) EOTB + MON.

The EOTB (ANAVRIN^®^), was provided at 0.35 g/100 kg body weight, while monensin (Rumensin^®^, Elanco, Krebz, Uruguay) was supplied at 0.033 g/kg dry matter of TMR. The dosage and inclusion criteria for each additive incorporated into the TMR were determined in accordance with the manufacturer’s specifications, as these recommendations represent the standard practice adopted by farmers. For the combination treatment (EOTB + MON), each additive was provided at the same individual dose as in 1 and 2 treatments.

Four steers per treatment had a rumen permanent catheter (8 mm diameter) inserted through the dorsal sac for rumen liquor sampling.

The TMR (Table 1) was formulated using the Beef Cattle Nutrient Requirements Model (2016, Version 1.0.37.15) software [24] to achieve a target average daily gain (ADG) of 1.4 kg/day per animal. The TMR was prepared daily (08:00–10:00 h) using a vertical mixer (Mary S.R.L, Santa Catalina, Uruguay), weighed using a floor scale (EL-5 Marvic Ltd., Montevideo, Uruguay), and individually supplied for each steer. The steers were fed at 10:00 and 16:00 h, providing approximately half of the daily allowance for each meal. The TMR offered for each steer was adjusted every 12 days based on BW gain. The additives were added to the TMR of each animal according to the mentioned doses.

The experiment lasted 60 days, including a 19-day adaptation period. Three 6-day periods for data collection and sampling were included, separated by 9-day intervals. Steers were housed in individual 15 m^2^ open-air pens with shade, feeders, and water available ad libitum.

### 2.2. Dry Matter Intake, Average Daily Gain, and Feed Conversion Efficiency

Individual dry matter intake (DMI) of TMR was measured over 5 consecutive days of each period, weighing the feed offered and refused using a scale EL-5 Marvic^®^ of 150 × 0.05 kg, Marvic ltda, Montevideo—Uruguay. Samples of the TMR offered and refused were taken each day and frozen at −20 °C for further analysis.

Steer body weights were determined by averaging two consecutive daily measurements taken at days 0–1, 14–15, 27–28, 40–41, and 54–55. All weightings were conducted before TMR feeding (08:00–10:00 h), using a bovine scale (Terko, Tk3515c, Montevideo, Uruguay). Individual daily gain (DG, kg/d) for each steer was calculated as the difference in weight (kg) divided by the number of days between weightings. Feed conversion ratio (FCR) was calculated as the ratio of DMI to the average DG (ADG) for the whole period.

### 2.3. Ruminal Environment

On the first day of each measurement period, approximately 50 mL rumen fluid samples were collected via a rumen catheter at eight time points (09:30, 13:00, 15:00, 17:00, 19:00, 21:00, 03:00, and 09:30 h) to determine the pH, NH_3_-N, and VFA concentrations. Rumen pH was measured immediately using a digital pH meter (EW-05991-36, Cole Parmer, Vernon Hills, IL, USA). A rumen fluid subsample (1 mL) was preserved with 0.02 mL of 50% (*v*/*v*) sulfuric acid and another one with 1 mL of 0.1 M perchloric acid, and frozen at −20 °C for subsequent NH_3_-N and VFA analysis, respectively. The NH_3_-N concentrations were determined colorimetrically using a spectrophotometer (1200, UNICO^®^, United Products & Instruments Inc., Dayton, OH, USA) and a phenol–hypochlorite reaction [25]. The VFAs (acetic, propionic and butyric) were analyzed by High-Performance Liquid Chromatography (HPLC; Dionex Ultimate 3000, Sunnyvale, CA, USA) according to Adams et al. [26], using an Acclaim Rezex Organic Acid H+ (8%) column (7.8 × 300 mm) set to 210 nm. Total VFA concentration was calculated as the sum of acetic, propionic, and butyric acid concentrations.

### 2.4. Chemical Analysis of Feeds

Offered and refused feed samples were dried in a forced-air oven at 60 °C, ground using a 1 mm sieve mill (Arthur H. Thomas Co., Philadelphia, PA, USA), and analyzed for dry matter (DM) and ash content [27] (Methods 942.05 and 934.01, respectively). Organic matter (OM) was calculated as the difference between DM and ash. Total N was determined using the Kjeldahl method [27] (Method 984.13), and crude protein (CP) was calculated as N × 6.25. Neutral detergent fiber (NDF) with α-amylase and sodium sulfite, as well as acid detergent fiber (ADF) [28], were determined, and the values presented include residual ash.

### 2.5. Statistical Analysis

Data were analyzed using SAS version 9.0 (SAS Institute Inc., Cary, NC, USA). Outliers were identified using the PROC UNIVARIATE. The variables DMI, weight gain, and DG, data were analyzed by the PROC MIXED using the following model:Yik = µ + Bi + Tj + Pk + Tj × Pk + eijk
where Yijk = dependent variable (DMI, weight gain, and DG); µ = overall mean; Bi = random effect of block (i = 1 to 5 BW blocks); Tj = fixed effect of treatment (j = EOTB, MON, EOTB + MON); Pk = random effect of day of measurement for DMI or FCR (k = 1 to 3) and random effect of the period for BW or DG (k = 1 to 5); Tj × Pk = fixed effect of the interaction between treatment and period; eijk = residual error.

For ADG, FCR only the random effect of the block and the fixed effect of treatment were included in the model, as follows:Yik = µ + Bi + Tj + eijk

The variables pH, NH_3_-N, and VFAs were analyzed as repeated measures over time using the following model:Yijkl = µ + Bi + Tj + Pk + Tj × Pk + Hl + Tj × Hl + eijkl
where Yijkl = dependent variables (pH, NH3-N, and VFAs); µ = overall mean; Bi = random effect of block (i = 1 to 5 BW blocks); Tj = fixed effect of treatment (j = EOTB, MON, EOTB + MON); Pk = random effect of the period (k = 1 to 3); Tj × Pk = fixed effect of the interaction between treatment and period; Hl = the fixed effects of hour of measurement (Hl = 9:30, 13:00, 15:00, 17:00, 19:00, 21:00, 03:00 h); Tj × Hl = interaction between treatment and hour; and eijk = residual error.

A spatial power (SP(POW)) for irregularly spaced data and Tukey’s test were used for mean separation, with significance declared at *p* < 0.05.

## 3. Results

The EOTB treatment led to higher DMI compared to EOTB + MON (*p* = 0.02) and this was the only difference in intake and performance (Table 2, Figure 1).

No differences were observed between treatments for average daily gain (ADG) or feed conversion ratio (FCR), as shown in Table 2.

The rumen fermentation variables (Table 3) showed little variation among treatments. No significant differences were observed in pH, NH3-N, and total VFAs. However, a significant treatment × hour interaction was observed for total VFAs, acetic acid, and butyric acid concentrations (expressed in Mm). These interactions are displayed in Figure 2. The VFA concentration was higher (*p* < 0.05) for EOTB and MON compared to EOTB + MON at hour 8 of measurement. Similar behavior was observed for acetic and butyric acids. At hour 24, VFA concentration was lowest for EOTB (*p* < 0.05), and a similar behavior was observed for propionic acid, while butyric showed the highest values for MON. Between treatments, there were differences in data expressed both as percentages and mM. Butyric acid was higher for MON compared to EOTB + MON (*p* = 0.01 expressed as mM, and *p* = 0.02 expressed as percentage). A tendency towards a higher proportion of propionic acid was observed in the EOTB + MON group compared to the EOTB group (*p* = 0.06).

## 4. Discussion

It is worth noting that this study did not include control treatment without additives. This was mainly because we had a limited number of animals for the experiment, and the team decided to focus on directly comparing the new additive to monensin, since there were no existing data on that comparison. While this study design does not allow for isolating the impact of each additive on productive performance, it offers the innovative advantage of enabling a direct comparison of the two additives and their combined effects.

Only minor differences in voluntary intake were observed among treatments. Similarly, Diepersloot et al. [29], working with high-producing dairy cows, reported no difference in DMI after adding EOs, monensin, or a combination of both for 10 weeks. However, in the aforementioned study, the additive’s effect on intake varied by week. According to Wood et al. [30], the intake reduction triggered by monensin is expected to be dose-dependent. These authors studied the effect of increasing the dose of monensin in crossbreed finishing heifers and observed a linear decrease of DMI as the dose increased from 0 to 48 mg monensin/kg of TMR. The highest dose employed by those authors was equivalent to 60 mg/kg DM, while in our study, monensin was provided at 33 mg/kg DM. Although the effect of monensin in reducing DMI is well documented [30,31], there is less information available regarding the impact of EO blends. In a recent study, Silvestre et al. [32], working with dairy cows, did not find an effect on DMI after adding EOs from geranium and cloves to the diet, while Atzori et al. [33] did not find differences between a control diet and the same diet supplemented with the blend of EOs, tannins, and bioflavonoids used in this article, in dairy sheep. This aligns with the findings of the metanalyses by Belanche et al. [9], who indicated that the essential oil blend had no impact on DMI or milk composition. The only difference observed in the present study was a higher DMI in the EOTB treatment compared to EOTB + MON, which suggests that the combination of EOTB and MON may have an adverse effect on DMI. In our study, the inclusion of tannins does not appear to have negatively affected intake, as reported in the literature [34], since the EOTB treatment had the highest absolute DMI value, and monensin does not contain tannins.

The higher DMI in the EOTB treatment did not affect ADG or FCR, which showed no differences between treatments. Although the absolute values could suggest a higher ADG with the EOTB treatments, the absence of differences indicates that more animals would be needed to confirm this assumption.

The rumen pH values observed in this study were comparable to those reported for similar diets and remained above the levels deemed at risk [35]. The similarity in rumen pH among treatments is consistent with the absence of differences in VFA concentrations. Neither Diepersloot et al. [29] nor Flores et al. [36] observed major differences in pH or VFA concentrations. The only notable finding reported by the latter was an increase in butyrate concentration. It is noteworthy that although tannins are generally recognized as VFA concentration reducers [37], in the EOTB treatment, VFAs were not reduced. This can be due to the type of tannins contained in this specific blend (from chestnut). Buccioni et al. [38], studying the effect of adding chestnut or quebracho tannins to dairy sheep diets, reported that tannins from quebracho reduced VFA concentrations, but tannins form chestnuts increased their concentration compared to the control. Among VFA, the higher butyric percentage observed with monensin was the main change observed. This can be related with the fact that, although monensin is known to be effective on Gram-positive bacteria, this observation is not applicable to all Gram-positive bacteria in the rumen. For example, *Butyrivibrio fibrisolvens*, a butyric acid-producing Gram-positive bacterium, is insensitive to dietary ionophores, apparently due to its morphological characteristics [3].

The interaction observed in VFA kinetics over time is an interesting finding. While a more stable VFA concentration for the EOTB + MON treatment cannot be definitively confirmed, it was evident that this treatment avoided the high peaks observed in the other treatments between 6 and 8 h post-feeding. Additionally, the interaction revealed that the butyrate concentration at 24 h was higher for the MON treatment compared to the others. Although these effects are subtle, they may indicate a potential action of monensin and the EOTB blend. Further in-depth studies on the rumen environment and microbiota are necessary to confirm these observations and elucidate the underlying mechanisms. It is known that EOs, tannins, and bioflavonoids can modify rumen microbiota [6,39,40]. However, the diversity of components and their specific actions make it difficult to identify the precise mechanisms at work in this situation.

The lack of differences in ruminal ammonia concentrations was expected. Monensin is known to inhibit amino acid-fermenting bacteria [41]. The blend of EOTB, on the other hand, contains tannins, which are known to reduce protein degradation by forming insoluble complexes with proteins [42]. Therefore, although through different mechanisms, both additives tend to reduce ruminal ammonia concentrations. The different mechanisms of both additives suggest potential complementarity, leading to the expectation that the EOTB + MON treatment would result in lower ammonium concentrations, which was not observed.

## 5. Conclusions

The addition of a blend of essential oils, tannins, and bioflavonoids to the diet of finishing steers fed a total mixed ration resulted in similar daily gain and feed conversion outcomes compared to monensin. Ruminal fermentation variables were also similar among treatments, except for a lower butyric concentration when the combination of both additives was used. This finding, along with the interactions observed in the fermentation kinetics between additives, can supports the need for further studies on the ruminal microbiome to better interpret the effects of this type of additive.

## Figures and Tables

**Figure 1 animals-15-00594-f001:**
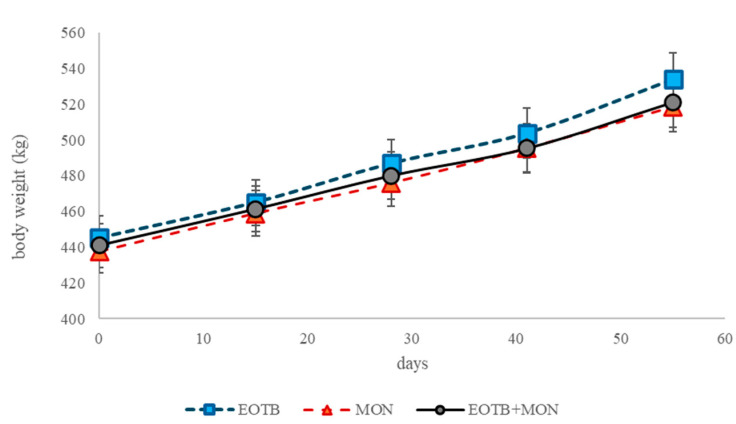
Body weight evolution during the last 60 days of fattening of steers fed a total mixed ration and supplemented with a blend of essential oils, tannins and bioflavonoids (EOTB), monensin (MON), or their combination (EOTB + MON). The bars represent the standard error of the mean.

**Figure 2 animals-15-00594-f002:**
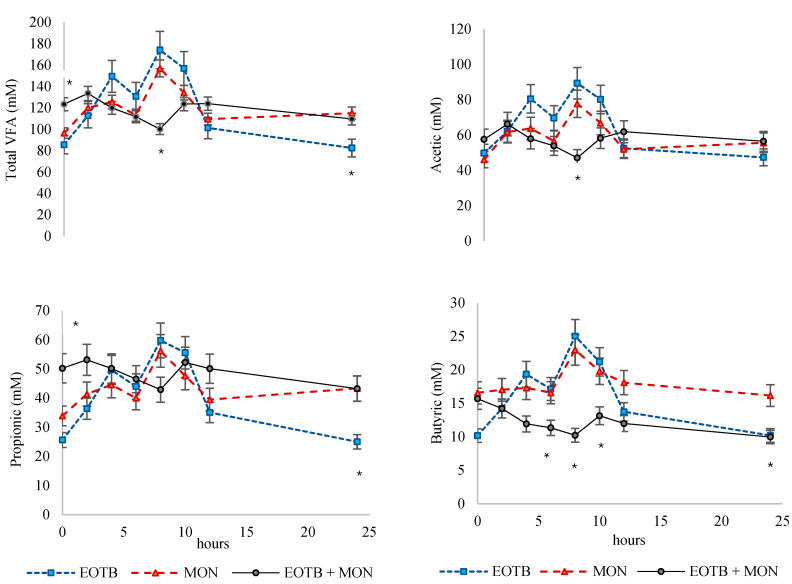
Daily kinetics od ruminal volatile fatty acids (VFAs) in steers fed a total mixed ration and supplemented with a blend of essential oils, tannins and bioflavonoids (EOTB), monensin (MON), or their combination (EOTB + MON). Each point corresponds to the average of 4 steers across 3 periods, * denotes at least one difference between treatments at *p* < 0.05, and the bars represent the standard error of the mean.

**Table 1 animals-15-00594-t001:** Proportion of each component of the total mixed ration (TMR, % of DM) and chemical composition of feeds used (mean values and SD in parenthesis).

	TMR	GCG	SM	Hay
Ingredients of TMR				
Ground corn grain (GCG)	71.7			
Solvent extracted soybean meal (SM)	10.1			
Pasture hay (Hay)	17.0			
Vitamin-mineral premix ^1^	1.2			
Water (% as feed)	27.1			
Nutrient composition				
DM, % as feed	88.3 (1.6)	86.7 (0.4)	91.3 (0.2)	86.6 (1.0)
OM	95.8 (0.4)	98.6 (0.01)	93.0 (0.01)	92.4 (0.01)
NDF	16.9 (2.8)	8.0 (0.4)	10.7 (0.4)	59.4 (1.0)
ADF	8.9 (0.8)	2.4 (0.1)	7.8 (0.9)	37.9 (0.6)
CP	11.4 (1.5)	9.0 (0.1)	38.6 (3.2)	6.3 (0.1)

^1^ Vitamin–mineral premix (Agrifirm S.A., Canelones, Uruguay): vitamin A, 53,000 UI; vitamin D, 10,600 UI; vitamin E, 200 UI; Co, 2.6 mg/kg; I, 18.4 mg/kg; Se inorg., 4.4 mg/kg; Zn inorg., 1200 mg/kg; Cu inorg., 421 mg/kg; Na, 6.3 g/kg; Mg, 1.5 g/kg; Ca, 22.0 g/kg.

**Table 2 animals-15-00594-t002:** Intake and performance of steers fed a total mixed ration and supplemented with a blend of essential oils, tannins and bioflavonoids (EOTB), monensin (MON), or their combination (EOTB + MON).

	EOTB	MON	EOTB + MON	SEM	*p*
DMI, kg/d	12.3 ^a^	11.8 ^ab^	11.3 ^b^	0.46	0.02
Initial BW, kg	447.1	440.9	435.9	11.50	0.62
Final BW, kg	534.7	519.8	516.3	15.26	0.33
Average DG, kg/d (60 d)	1.5	1.3	1.4	0.10	0.40
FCR, DMI/DG	8.7	9.1	8.9	0.66	0.90

DMI, dry matter intake; BW, body weight; DG, daily gain; FCR, feed conversion ratio; SEM, standard error of the mean; ^a,b^ values with different superscript within the same row are different (*p* < 0.05).

**Table 3 animals-15-00594-t003:** Rumen environment of steers fed a total mixed ration and supplemented with a blend of essential oils, tannins and bioflavonoids (EOTB), monensin (MON), or their combination (EOTB + MON).

					*p*		
	EOTB	MON	EOTB + MON	SEM	T	H	T × H
pH	6.0	5.8	5.9	0.06	0.18	<0.01	0.94
Total VFAs, mM	127.3	123.4	119.4	8.80	0.75	<0.01	0.03
Acetic, mM	67.0	63.6	56.3	5.39	0.15	<0.01	0.05
Propionic, mM	44.2	42.6	49.7	5.90	0.50	<0.01	0.11
Butyric, mM	16.2 ^ab^	18.4 ^a^	14.4 ^b^	1.77	0.01	<0.01	0.01
Acetic, %	53.7	50.1	47.4	3.51	0.37	<0.01	0.10
Propionic, %	33.3 ^y^	34.9 ^x, y^	42.1 ^x^	3.27	0.06	<0.01	0.84
Butyric, %	13.0 ^ab^	14.7 ^a^	11.5 ^b^	1.42	0.02	0.21	0.18
Acetic/Propionic	1.7	1.5	1.1	0.26	0.18	<0.01	0.81
NH_3_-N_,_ mg/dL	17.9	16.9	20.7	8.20	0.684	<0.01	0.90

VFAs, volatile fatty acids; SEM, standard error of the mean; T, treatment; H, hour; T × H, interaction between treatment and hour; ^a,b^ values with different superscript within the same row are significantly different for the treatment effect at *p* < 0.05; ^x,y^ values with different superscript within the same row indicate tendencies toward differences in the treatment effect (0.05 < *p* < 0.1).

## Data Availability

The data presented in this study are available on request from the corresponding author due to research agreement restrictions.

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
