# Peer review of "Daily Gain, Feed Conversion, and Rumen Fermentation in Finishing Steers Fed a Total Mixed Ration Supplemented with a Blend of Essential Oils, Tannins, and Bioflavonoids or Monensin†"

_animals, 2025, doi:10.3390/ani15040594_

Round 1

Reviewer 1 Report

Comments and Suggestions for Authors

Dear Authors,

congratulations on taking up an important topic. Nevertheless, the work submitted for review contains some defects that do not allow its acceptance in its current form. The comments concern primarily the methodology, which requires some explanations and systematization. Some of the discussed results are not reflected in the presented tables and graphs, and others remain controversial. I hope that the comments found in the attached file will allow for the improvement of your manuscript.

Yours sincerely.

Author Response

Reviewer 1

Dear Authors,

congratulations on taking up an important topic. Nevertheless, the work submitted for review contains some defects that do not allow its acceptance in its current form. The comments concern primarily the methodology, which requires some explanations and systematization. Some of the discussed results are not reflected in the presented tables and graphs, and others remain controversial. I hope that the comments found in the attached file will allow for the improvement of your manuscript.

Yours sincerely.

AU: Thank you very much for your positive comments and for the detailed revision, which was very useful for us to improve the manuscript.  We tried to improve the description and systematization of the methodology, and we reviewed several aspects of the discussion and presentation of the results.

We want to point out that we realized we had made a mistake in Figure 2, leading to comments. The x-axis of the figure corresponds to hours, not to days. We apologize for the difficulties this fact would have led to during the review.

We responded to each comment in the same PDF document you sent.

Reviewer 2 Report

Comments and Suggestions for Authors

The manuscript is clear, relevant for the field and presented in a well-structured manner. It provides evidence based results as it was performed following an appropriate experimental design to reach the objective. Their objective was to evaluate the effects of a specific blend of essential oils, tannins, and bioflavonoids compared to monensin and a combination of both, on the performance and rumen fermentation parameters of finishing steers consuming a total mixed ration. The simple fact that they could prove a similar animal performance with the blend compared to monensin is interesting, as it allows the producer to avoid antimicrobial use. On the other hand, the hint of a possible synergia between both additives opens the window for future studies. 

In general, their results could be reproduced based on the details given in the manuscript. It would have been valuable to have a control group, so they could compare different additives against no additive also. Although in general, the methodology is explained, they have some inaccuracies and lack of detail. 

Generally, the results are properly presented in tables and figures, they are easy to understand.  

Although in the figure you can see different patterns in VFA concentrations among time, I don’t think there is enough evidence to say a treatment has a more stable VFA concentration and to conclude about it. I recommend making an appropriate analysis to have this kind of results or stating this as a potential.

The introduction gives a general view of the state of the art about the use of monensin around the globe and the addition of phytogenic compounds to promote rumen health. They identified the gap in knowledge on the effects of the combination of phytogenic compounds with monensin. Although most of the references are more than 10 years old they are relevant.

Specific comments

  1. In the introduction, in line 77, they state “monensin showed clearer effects than EOs [10]”, please specify what clearer effects means. 
  2. At the end of the introduction, the objective of the study is duplicated. 
  3. In methodology, I would like to have a better explanation of the different periods within the study (line 138) to appropriately interpret the graphs. And the length of the adaptation period is different in 2 parts of the manuscript: 14 d (line 24) or 19 d (line 138).
  4. They should always refer to ammonia nitrogen as NH3-N; sometimes they say NH3 and sometimes N-NH3 or NH3-N. 
  5. I would like to read how the additives were fed. Inside the premix? If so, it should also be stated in table 1 (they only refer to anavrin).
  6. In table 1 the title should say the means are shown. Some decimal numbers are separated with comas and some with dots. Thousands should have comas when explaining the composition of the premix. The digits of decimal numbers should be the same for the same variable. For OM, a feed shows only 1 decimal and others show 3 decimals in the SD. For NDF it is similar for the mean.  
  7. In line 184, it says K goes from 1 to 3 or from 1 to 5, please explain further.
  8. Title of figure 1 should refer to the period of the study. 
  9. Table 2. I think there is no sense in stating “x,y values with different superscript within the same row are tendencies (0.05<P<0.1)” as there are no cases in the table.
  10. In line 219, they should make reference to Table 2, were those results are shown.
  11. In table 3 there should be a superscript for the details given below the table.
  12. In figure 2, some Y axis names have there units between parenthesis, and some not. In its title they should make reference to the period (X axis) of the study.  
  13. In line 256 they state:  the additive's effect on intake varied by week, but there are no results showing this.
  14. I don’t agree with the comment in line 271 about the counteracting effect of EOTB. If so, they would also have found a difference between EOTB and MON. The same for line 272. To say that they should have the results of a control group. 
  15. In the paragraph initiated in line 274, they could also state that there were differences in initial BW that could have contributed to the differences in DMI or ADG.

Author Response

Reviewer 2

Comments and Suggestions for Authors

The manuscript is clear, relevant for the field and presented in a well-structured manner. It provides evidence based results as it was performed following an appropriate experimental design to reach the objective. Their objective was to evaluate the effects of a specific blend of essential oils, tannins, and bioflavonoids compared to monensin and a combination of both, on the performance and rumen fermentation parameters of finishing steers consuming a total mixed ration. The simple fact that they could prove a similar animal performance with the blend compared to monensin is interesting, as it allows the producer to avoid antimicrobial use. On the other hand, the hint of a possible synergia between both additives opens the window for future studies. 

 In general, their results could be reproduced based on the details given in the manuscript. It would have been valuable to have a control group, so they could compare different additives against no additive also. Although in general, the methodology is explained, they have some inaccuracies and lack of detail. 

Generally, the results are properly presented in tables and figures, they are easy to understand.  

Although in the figure you can see different patterns in VFA concentrations among time, I don’t think there is enough evidence to say a treatment has a more stable VFA concentration and to conclude about it. I recommend making an appropriate analysis to have this kind of results or stating this as a potential.

The introduction gives a general view of the state of the art about the use of monensin around the globe and the addition of phytogenic compounds to promote rumen health. They identified the gap in knowledge on the effects of the combination of phytogenic compounds with monensin. Although most of the references are more than 10 years old they are relevant.

AU: Thank you very much for your positive comments about our study, and the careful revision of the manuscript, which will help us to improve the final result.

We tried to improve the manuscript and follow the recommendations provided. First of all, we sincerely apologize for the error in Figure 2. Instead of labeling the X-axis with "hours," as it should have been, we mistakenly labeled it as "days." This fact confused not only in understanding the methodology but also in interpreting the results. We also have reviewed certain aspects of the discussion and presentation of the results.

  1. In the introduction, in line 77, they state “monensin showed clearer effects than EOs [10]”, please specify what clearer effects means. 

AU: we added information about the paper cited.

  1. At the end of the introduction, the objective of the study is duplicated. 

AU: Thank you, we combined both sentences.

  1. In methodology, I would like to have a better explanation of the different periods within the study (line 138) to appropriately interpret the graphs. And the length of the adaptation period is different in 2 parts of the manuscript: 14 d (line 24) or 19 d (line 138).

AU: Thanks for the observation. The periods were correctly explained, but the confusion is due to the error mentioned in figure 2. Concerning the length of the adaptation period, which was 19 days, we made a mistake in the simple summary and abstract. Sorry, we corrected both.

  1. They should always refer to ammonia nitrogen as NH3-N; sometimes they say NH3 and sometimes N-NH3 or NH3-N. 

AU: Thank you very much for your observation. We have corrected and standardized throughout the text.

  1. I would like to read how the additives were fed. Inside the premix? If so, it should also be stated in table 1 (they only refer to anavrin).

AU: Thank you for your comment. The additives were provided in the TMR mixture. They were added separately for each treatment. In table 1 there is an error. Although Vitamin-mineral premix (Agrifirm S.A., Uruguay) was used, on this occasion it did not have Anavrin as it was prepared for the experiment. We added the information in M and M and corrected the information in table 1.

  1. In table 1the title should say the means are shown. Some decimal numbers are separated with comas and some with dots. Thousands should have comas when explaining the composition of the premix. The digits of decimal numbers should be the same for the same variable. For OM, a feed shows only 1 decimal and others show 3 decimals in the SD. For NDF it is similar for the mean.  

AU: Thank you for your comment. Table 1 was corrected.

  1. In line 184, it says K goes from 1 to 3 or from 1 to 5, please explain further.

AU: Thank you, this was corrected

  1. Title of figure 1 should refer to the period of the study. 

AU: The period of study was added to the title.

  1. Table 2. I think there is no sense in stating “x,y values with different superscript within the same row are tendencies (0.05<P<0.1)” as there are no cases in the table.

AU: Yes, we agree, it was deleted

  1. In line 219, they should make reference to Table 2, were those results are shown.

AU: Added

  1. In table 3 there should be a superscript for the details given below the table.

AU: We are not sure to have understood the observation. We used the superscripts only for detail differences and tendencies.

  1. In figure 2, some Y axis names have there units between parenthesis, and some not. In its title they should make reference to the period (X axis) of the study.  

AU: Thanks, it was corrected.

  1. In line 256 they state:  the additive's effect on intake varied by week, but there are no results showing this.

AU: Sorry, we were referring to Diepersloot et al. study. We added this information to the text.

  1. I don’t agree with the comment in line 271 about the counteracting effect of EOTB. If so, they would also have found a difference between EOTB and MON. The same for line 272. To say that they should have the results of a control group. 

AU: Thank you, the first comment was corrected. In the second comment, we agree that a control group would be useful. However, the diet with monensin did not contain tannins. We changed the sentence to clarify our idea.

  1. In the paragraph initiated in line 274, they could also state that there were differences in initial BW that could have contributed to the differences in DMI or ADG.

AU: We do not understand the observation. There were no differences in initial BW (see table 2), which was not surprising as the animals were blocked by BW. Also, we could not detect differences in ADG. Please, if you think we should change the sentence, let us know.

Reviewer 3 Report

Comments and Suggestions for Authors

REVIEW REPORT FOR ANIMALS- 3456205 - SCIENTIFIC AND DETAILED COMMENTS

The reviewer appreciates and thanks the journal for the review process of Manuscript ID: Animals- 3456205, titled Performance and Rumen Fermentation in Finishing Steers Fed a Total Mixed Ration Supplemented with a Blend of Essential Oils, Tannins, and Bioflavonoids or Monensin” has been reviewed.

S/N

Comments

Reviewer 

1

Simple Summary

Is ok with minor critiques 

2

Abstract

Why was the unit of measurement for the inclusion of EOTB in g/100 kg body weight while that of MON is g/kg dry matter? Could the authors justify that differences?  

3

Introduction

The introduction is well-written with minor corrections 

4

Methodology

The procedures are quite explicit. 

5

What is the main question addressed by the research?

This research investigates the potential benefits of essential oil blends (EOs) on the performance and rumen fermentation of finishing steers fed a mixed ration. While the study demonstrated a positive effect of EOs on feed intake, the majority of other productive performance parameters remained unaffected. Similarly, a positive trend was observed for rumen fermentation of volatile fatty acids in the EO-supplemented group.

6

What specific improvements should the authors consider regarding the methodology? 

The study design is not explicitly stated in the methodology section. 

While the discussion acknowledges the absence of a control group without any additives, it is unclear how this design allows for a direct comparison of the two additives and their combined effect.

For clarification, the authors should explicitly state the experimental design employed in this study and provide a more detailed explanation of how the absence of a control group facilitates the intended comparisons.

7

Statistical analysis

The statistical tools employed for the analysis of different variables were correct 

8

Results

For clarification, please provide clearer differences between the results presented in Table 3 and Figure 2, as well as the Table 3 footnote.

9

General observations (originality of research topic and relevance to the field, address a specific gap in the field, depth of the research in the field, 

Yes, the topic is original and relevant to the field.

Yes, it bridges the knowledge gap by shedding light on the potential EO’s and their blend on the performance and rumen fermentation mechanisms of finishing steers, which has a paucity of scientific information particularly on cattle. However, there is a need for some additional studies especially the microbiome of post-treated EO’s animals.

10

Are the conclusions consistent with the evidence and arguments presented and do they address the main question posed? Please also explain why this is/is not the case.

The conclusion is Ok. Also, it is very crucial to further the microbiome profile studies.

11

Are the references appropriate?

Yes, the majority of the references listed are current and relevant. However, a few minor amendments may be required for some entries.

Author Response

Reviewer 3

REVIEW REPORT FOR ANIMALS- 3456205 - SCIENTIFIC AND DETAILED COMMENTS

The reviewer appreciates and thanks the journal for the review process of Manuscript ID: Animals- 3456205, titled “Performance and Rumen Fermentation in Finishing Steers Fed a Total Mixed Ration Supplemented with a Blend of Essential Oils, Tannins, and Bioflavonoids or Monensin” has been reviewed.

AU: Thank you very much for your positive contribution to this review. We want to apologize for an error in Figure 2. Instead of labeling the x-axis with "hours," we mistakenly labeled it as "days." This fact evidently confused the interpretation of the results.

We respond to each observation in the table below and also in the same pdf with comments. We hope we could improve the manuscript in the new version.

S/N

Comments

Reviewer 

1

Simple Summary

Is ok with minor critiques 

AU: Thank you. See corrections in the new version and responses in the pdf

2

Abstract

Why was the unit of measurement for the inclusion of EOTB in g/100 kg body weight while that of MON is g/kg dry matter? Could the authors justify that differences?  

AU: we followed the manufacturer's recommendations for each additive, as it is the way of use by farmers. We added this explanation in the M and M section

3

Introduction

The introduction is well-written with minor corrections 

AU: Thank you. See corrections in the new version and responses in the pdf

4

Methodology

The procedures are quite explicit. 

AU: Thank you. See corrections in the new version and responses in the pdf

5

What is the main question addressed by the research?

This research investigates the potential benefits of essential oil blends (EOs) on the performance and rumen fermentation of finishing steers fed a mixed ration. While the study demonstrated a positive effect of EOs on feed intake, the majority of other productive performance parameters remained unaffected. Similarly, a positive trend was observed for rumen fermentation of volatile fatty acids in the EO-supplemented group.

AU: Thank you

6

What specific improvements should the authors consider regarding the methodology? 

The study design is not explicitly stated in the methodology section. 

While the discussion acknowledges the absence of a control group without any additives, it is unclear how this design allows for a direct comparison of the two additives and their combined effect.

For clarification, the authors should explicitly state the experimental design employed in this study and provide a more detailed explanation of how the absence of a control group facilitates the intended comparisons.

AU: Undoubtedly, including a control group would have allowed for a more comprehensive discussion, and we fully acknowledge this limitation. This is precisely why we addressed this point at the beginning of the discussion section.

As with any controlled experiment, the number of animals available for the study was limited. Our primary objective was to ensure that each treatment group had a sufficient number of animals to detect potential differences, should they exist. Consequently, we had to make strategic decisions about which treatments to include in the experimental design.

Below, we outline the rationale for comparing the phytogenic additive with monensin, as well as the combination of both:

1.           Monensin has been extensively used for decades, with a well-documented background on its benefits for performance and rumen fermentation. For this reason, we considered monensin an appropriate positive control. As highlighted in the introduction, monensin is widely utilized in regions with significant beef cattle production.

2.           While the specific blend of essential oils (EOs), tannins, and bioflavonoids has been studied in comparison to control diets, it had not been previously evaluated against monensin. This comparison represents a novel contribution to the existing literature.

3.           The comparison between monensin, the phytogenic additive, and their combination has been a recurring request from farmers, reflecting its practical relevance and potential application in the field.

To provide further clarity, we have added a comment in the discussion section explicitly addressing the absence of a control group without additives.

7

Statistical analysis

The statistical tools employed for the analysis of different variables were correct 

AU: Thank you

8

Results

For clarification, please provide clearer differences between the results presented in Table 3 and Figure 2, as well as the Table 3 footnote.

AU: thank you for the contribution. The interpretation is clearer in your proposal. We re-wrote this paragraph taking into account your and the other reviewers’ suggestions.

9

General observations (originality of research topic and relevance to the field, address a specific gap in the field, depth of the research in the field, 

Yes, the topic is original and relevant to the field.

Yes, it bridges the knowledge gap by shedding light on the potential EO’s and their blend on the performance and rumen fermentation mechanisms of finishing steers, which has a paucity of scientific information particularly on cattle. However, there is a need for some additional studies especially the microbiome of post-treated EO’s animals.

AU: Thank you

10

Are the conclusions consistent with the evidence and arguments presented and do they address the main question posed? Please also explain why this is/is not the case.

The conclusion is Ok. Also, it is very crucial to further the microbiome profile studies.

AU: Thank you

11

Are the references appropriate?

Yes, the majority of the references listed are current and relevant. However, a few minor amendments may be required for some entries.

AU: Thank you for the detailed revision. See corrections in the new version and responses in the pdf

Round 2

Reviewer 1 Report

Comments and Suggestions for Authors

Dear Authors,

Thank you for taking into account the suggestions sent earlier. In my opinion, the manuscript in its current form is ready for publication. Once again, congratulations on choosing the topic and I wish you further research successes.
Yours sincerely